# Reciprocal Associations among Social–Emotional Competence, Interpersonal Relationships and Academic Achievements in Primary School

**DOI:** 10.3390/bs13110922

**Published:** 2023-11-12

**Authors:** Jiaqi Yang, Yuze Deng, Yehui Wang

**Affiliations:** Collaborative Innovation Center of Assessment for Basic Education Quality, Beijing Normal University, Beijing 100875, China; 202231630005@mail.bnu.edu.cn (J.Y.); 202221630001@mail.bnu.edu.cn (Y.D.)

**Keywords:** social–emotional competence, academic achievements, teacher–student relationships, social preference, social impact

## Abstract

The study aimed to examine the reciprocal associations among social–emotional competence (SEC), interpersonal relationships (including teacher–student relationships and peer relationships) and academic achievements in reading, mathematics and science of primary school students. The Chinese versions of the Delaware Social and Emotional Competency Scale, Teacher–student Relationship Scale, Peer-nomination method and Academic Achievement Tests were used to measure students’ SEC, teacher–student relationships, peer relationships (including social preference (SP) and social impact (SI)) and academic achievements, respectively. In total, 3995 fourth-grade students participated in the first survey, and 2789 of them were tracked in the follow-up survey two years later. Structural equation modelling was employed to investigate the cross-lagged relationships among the variables across time. The results showed the reciprocal associations between SEC with academic achievements and teacher–student relationships, as well as between academic achievements with SP and teacher–student relationships. Moreover, variations in these reciprocal associations were observed across the subjects of reading, mathematics and science. In summary, this study offers new insights for enhancing students’ SEC, interpersonal relationships and academic achievements, and implications for future subject-specific education can be derived by considering the complex interplay in the subjects of reading, mathematics and science.

## 1. Introduction

Students’ success in school is not only influenced by their cognitive abilities [1,2,3,4]. It is also influenced by students’ ability to manage goal- and task-oriented behaviors, maintain social relationships and regulate emotions [1,4,5]. These abilities are collectively referred to as social–emotional competence (SEC) [6,7,8,9]. The National Academy of Sciences reports that only 40% of children have the social–emotional skills needed to succeed. Students with a low level of SEC often encounter difficulties in social interactions with teachers and peers, resulting in diminished classroom relatedness and negative academic performance [10,11,12,13]. Based on the importance of SEC for children’s development, it has received widespread attention. 

Many social–emotional learning (SEL) programs aimed at improving students’ SEC have achieved significant benefits [14,15]. The successful experience indicates that the best learning and development of children come from supportive environments and positive interpersonal relationships [16]. Teacher–student relationships and peer relationships constitute the main interpersonal relationships in the school context, playing a crucial role in fostering their development of SEC and learning outcomes [17,18,19]. In the process of children’s growth, many aspects and elements are considered to have interactive effects [20]. It is believed that interpersonal relationships (teacher–student relationships and peer relationships) and children’s development in SEC and academic achievements might have bidirectional associations. 

However, previous studies have focused on a limited number of these factors [19,21,22,23]. In fact, comprehensively considering these factors has a much higher degree of ecological validity, and provides empirical evidence about how to achieve better interpersonal relationships and improve SEC and academic achievements among primary school students. Moreover, it is essential to recognize that students’ relationships with their teachers vary across subjects [24]. Therefore, this study aims to investigate the reciprocal associations among SEC, teacher–student relationships, peer relationships and academic achievements in the subjects of reading, mathematics and science among primary school students.

### 1.1. Social–Emotional Competence and Academic Achievements

The promotion of SEC positively impacts cognitive and academic performance across various educational stages, including preschool, primary school and high school [25]. Insufficient opportunities to acquire and develop SEC impede students’ success in school [14,26]. High SEC empowers students to manage their emotions better, tackle challenges, acquire new knowledge more effectively and eventually achieve academic goals [27]. Growing evidence highlights the interconnectedness between SEC and academic achievements in reading, mathematics and science [5,14,15,27,28]. And the significance of SEC in promoting students’ subsequent academic achievements is also underscored [26,29]. Therefore, there is compelling evidence to believe that students’ SEC is a predictor of their subsequent academic achievements in reading, mathematics and science.

However, there is a lack of research on the impacts of students’ academic achievements on their subsequent SEC. According to Hattie and Timperley [30], academic achievements serve as feedback, providing students with valuable insights into their performance and understanding. In line with this, academic achievements may trigger a series of processes, such as judging one’s behavior based on personal standards, environmental conditions and emotional self-reflection. Empirical studies have already found that this feedback affects academic self-concept [31] and emotions [32]. Thus, it is reasonable to suggest that academic achievements are positive predictors of SEC. 

Based on these previous studies, the following hypothesis was designed for this study: 

**Hypothesis** **1** **(H1).***SEC at Time 1 (T1) will positively predict academic achievements in reading, mathematics and science at Time 2 (T2). Furthermore, academic achievements in reading, mathematics and science at T1 will positively predict SEC at T2*.

### 1.2. Social–Emotional Competence and Interpersonal Relationships

Interpersonal relationships, specifically teacher–student relationships and peer relationships, act as the basis for students’ development of SEC [33]. Longitudinal research has demonstrated that positive teacher–student relationships predict the development of essential SEC skills, such as self-control and interpersonal skills over time [11]. The reciprocal association between SEC and teacher–student relationships was investigated by Hajovsky et al. [11], highlighting the importance of strong interpersonal skills in the establishment and maintenance of close teacher–student relationships.

Furthermore, peer relationships also significantly impact social and emotional competence development [10]. Researchers have advocated thinking of different forms of peer relationships [34,35]. Although those who are genuinely well-liked by their peers and those who are seen as popular but are not necessarily well-liked are both popular outwardly, there is an internal difference between them. Social preference (SP) and social impact (SI) are a set of indicators of peer relationships, denoting the two different types of popularity. SP results from subtracting children’s liking nominations from their disliking nominations, representing the level to which they are genuinely well-liked by peers. SI is the sum of nominations as liked and disliked, reflecting the degree of being neglected (low score on SI) or controversial (high score on SI) among peers [36]. 

The degree of SP among peers plays a role in shaping children’s social competence, as a higher level of SP is more related to the use of multiple prosocial strategies [37]. Conversely, encountering difficulties in peer relationships can lead to subsequent aggressive behaviors [38]. Zimmer-Gembeck et al. [35] found a positive prediction of SI on subsequent aggressive behaviors. They also found a negative predictive effect of SP on subsequent aggressive behaviors, and a positive effect of SP on later prosocial behaviors. Longitudinal studies have shown that children who employ multiple social strategies are more likely to be well-liked by their peers [39,40]. Conversely, those with low social and emotional competence face disadvantages within the classroom [41]. 

Based on these previous studies, the following hypotheses were formulated for this study: 

**Hypothesis** **2** **(H2).***SEC at T1 will positively predict teacher–student relationships in reading, mathematics and science at T2. And teacher–student relationships in reading, mathematics and science at T1 will positively predict SEC at T2*.

**Hypothesis** **3** **(H3).***SEC at T1 will positively predict SP and negatively predict SI at T2. And SP and SI at T1 will predict SEC positively and negatively at T2, respectively*. 

### 1.3. Interpersonal Relationships and Academic Achievements

A meta-analysis revealed significant correlations between academic achievements and different teacher–student relationships (positive: *r* = 0.14; negative: *r* = −0.19) in primary school [42]. Establishing positive relationships with teachers has numerous academic benefits for students, including receiving constructive guidance, increased learning opportunities [17], and improved academic achievements [27]. A longitudinal study demonstrated that prolonged exposure to disharmonious teacher–student relationships increased the likelihood of academic failure [43]. Additionally, Aluja-Fabregat et al. [44] revealed that students’ prior academic achievements influenced teachers’ perceptions of them. Usually high-achieving students will receive greater preference from their teachers. 

Peer relationships also play a significant role in students’ academic achievements. A longitudinal study showed that negative peer relationships increased the risk of decreased academic achievements, whereas positive peer relationships positively predicted subsequent academic achievements [19,45]. Specifically, low SP was associated with declining academic achievements among primary school students [23,46], while a higher level of peer preference was linked to high academic achievements [33,46]. Furthermore, it is found that compared to individuals with lower academic performance, higher academic achievers engage in more positive interactions with peers and also obtain more acceptance, and usually have greater visibility or SI [47,48]. 

Based on these previous studies, the following hypotheses were proposed for this study: 

**Hypothesis** **4** **(H4).***Teacher–student relationships in reading, mathematics and science at T1 will positively predict academic achievements in the corresponding subject at T2. And academic achievements in reading, mathematics and science at T1 will positively predict teacher–student relationships in the corresponding subject at T2*. 

**Hypothesis** **5** **(H5).***SP and SI at T1 will predict academic achievements in reading, mathematics and science positively and negatively at T2, respectively. And academic achievements in reading, mathematics and science at T1 will positively predict SP and negatively predict SI at T2*. 

## 2. Materials and Methods

### 2.1. Participants and Procedure

In line with the purpose of the research, the current study adopted a two-wave longitudinal panel design, collecting data on the same variables with the same procedure at the first survey (T1) and the second survey (T2). Primary schools in four Chinese provinces were stratified based on their location (urban or rural areas). Participating students were selected using the whole class sampling method, with one to two classes sampled from each selected school. A total of 3995 fourth-grade students (47.5% girls; age: *M* = 10.76 years, *SD* = 0.90) participated in the first survey (T1). At T2, 2789 of them (46.8% girls, age: *M* = 12.43 years, *SD* = 1.18) were tracked two years later. There were 1206 participants missing due to school mergers, students changing schools and illness.

At both times, the participating students completed three academic achievement tests and background questionnaires on paper and pencil with the assistance of the teachers. The background questionnaires included measures of individual and family information. It took approximately 40 min for students to complete the reading, mathematics and science achievement tests, respectively, and almost 20 min to finish the background questionnaires. Informed consent was obtained from the principals, parents and students at the participating schools. All the participating students were informed of the voluntary and anonymous nature of the study. They could withdraw from the study at any time at their or their parents’ will.

### 2.2. Measures

Social–emotional competence was measured by the Chinese version of the Delaware Social and Emotional Competency Scale (DSECS-SCV) [49]. There were four dimensions: responsible decision making (example item, “I blame others when I’m in trouble”), relationship skills (example item, “I am good at solving conflicts with others”), self-management (example item, “I think before I act”) and social awareness (example item, “I care about how others feel”). These dimensions are each measured with 3 items, with a 4-point Likert scale from 1 = not like me at all to 5 = very much like me. The Cronbach’s α coefficient of the DSECS-SCV was 0.78. And the results of confirmatory factor analysis (CFA) indicated that the data fitted the model well (χ^2^/*df* = 3.516; CFI = 0.938; TLI = 0.918; RMSEA = 0.048) [49]. In this study, reliability analysis showed that Cronbach’s α coefficients for the scale were 0.82 at T1 and 0.83 at T2.

Teacher–student relationships were measured by the teacher–student relationship scale, translated into Chinese from the Programme for International Student Assessment (PISA) 2012 [50]. The wording was modified to make these items suitable for fourth-grade and sixth-grade students in China. Teacher–student relationships in the three subjects (reading, mathematics and science) were measured by 5 items (example item, “The teacher really listens to what I have to say”). Students rated each item on a 4-point Likert scale from 1 = strongly disagree to 4 = strongly agree. In the study, the scale had good reliability, with Cronbach’s α coefficients of 0.83, 0.82, 0.83 at T1, and 0.89, 0.91, 0.93 at T2 for reading, mathematics and science, respectively. The CFA results indicated that the data fitted the model well in the current study (χ^2^/*df* = 42.62~60.48; CFI = 0.982~0.989; TLI = 0.964~0.979; RMSEA = 0.045~0.054; SRMR = 0.017~0.022 at T1 and χ^2^/*df* = 36.48~60.48; CFI = 0.982~0.991; TLI = 0.964~0.981; RMSEA = 0.048~0.056; SRMR = 0.013~0.022 at T2).

Peer relationships were measured using the peer-nomination method [51]. Each child had a name list of their classmates and was asked to choose three favorites and three least favorites. Then, the numbers of each student being liked and disliked were standardized within the class to obtain their standardized liking (L) and disliking (D) scores. Subtracting the standardized D score from the standardized L score gives the indicator of SP, while the sum of the two scores gives the indicator of SI [51].

Academic achievements in reading, mathematics and science were measured using the reading, mathematics and science academic achievement tests, respectively, which were developed collaboratively by teachers and experts in educational measurement in accordance with the curriculum standards [42,43,46]. The reading, mathematics and science academic achievement tests contained 37, 41 and 40 items, respectively. The Rasch model was used to estimate students’ academic achievements with Conquest 3.0 [52]. This allowed for incorporating measurement errors and provided a more comprehensive understanding of students’ latent academic abilities [53,54]. The achievement scores were converted to a scale of 0–100. In this study, the three tests demonstrated good internal consistency, with Cronbach’s α coefficients of 0.75, 0.76, 0.76 at T1 and 0.83, 0.89, 0.82 at T2 for reading, mathematics and science, respectively.

### 2.3. Data Analysis

Analyses were conducted with IBM SPSS Version 24 and Mplus 8.3. First, Harman’s single-factor test was used to control the potential common method bias. Then, descriptive statistics, zero-order correlations, reliability analysis and analysis of covariance (ANCOVA) were performed in SPSS 24. Zero-order correlations were used to examine the associations between variables, and reliability analysis was used to assess the internal consistency (Cronbach’s alpha coefficient in the current study) of the scales. ANCOVA was employed to explore the impacts of covariates on the differences between the previous level and subsequent level of variables after controlling for the corresponding variables at the previous level. 

Second, an independent *t*-test was conducted to assess the problem of longitudinal attrition. Specifically, the significance of differences in the variables studied in the study between the completers and noncompleters was estimated. 

Third, the measurement invariance by time for the latent variables (SEC and teacher–student relationships in reading, mathematics and science) were examined using Mplus 8.3 [55]. This underpins the test of autoregressive and cross-lagged effects of the variables. Four nested models (configural, metric, scalar and residual invariance models) were examined by sequentially adding constraint equality conditions (configurations, factor loading, intercept and residual variance) to the less strictly limited model to conduct measurement invariance tests at different levels [56]. The chi-square test (χ^2^) is one of the most common indicators of the measurement invariance test but is sensitive to sample size [57]. The differences in the goodness-of-fit indices (comparative fit index (CFI), Tucker—Lewis index (TLI), root mean square error of approximation (RMSEA), and standardized root mean square residual (SRMR) were calculated by subtracting the fit indices of the less strictly limited model from those of the more strictly limited model. Absolute differences smaller than 0.01 indicate that the limited equivalence does not weaken the model fit; absolute differences between 0.01 and 0.02 indicate a moderate difference; and absolute differences larger than 0.02 indicate that the difference is obvious [58,59]. 

The reciprocal associations among students’ SEC, SP, SI, teacher–student relationships and academic achievements in reading, mathematics and science were explored separately using Mplus 8.3 [55], controlling students’ gender, age and their parents’ highest education. The robust maximum likelihood estimator (MLR) was used to estimate the results. The same indices of goodness-of-fit for the models as in the measurement invariance tests were used here. Values of CFI and TLI higher than 0.95 reflect a good fit between the model and data, while values between 0.90 and 0.95 are considered acceptable. In addition, an RMSEA value lower than 0.06 shows a good model fit between the model and data, while a value between 0.06 and 0.08 is regarded as acceptable [60]. Moreover, an SRMR value of less than 0.05 is considered a good fit between the model and data [61].

## 3. Results

### 3.1. Common Method Bias Test 

The results of Harman’s single-factor test showed that there were nine factors with eigenvalues higher than 1. Nearly 22.73% of the variance was explained by the first factor, lower than the critical value of 50% [62], suggesting that there was no serious common method bias. 

### 3.2. Descriptive Statistics, Correlations and ANCOVA

The descriptive statistics results showed the means and standard deviations of SEC (*M* = 3.01/3.19, *SD* = 0.55/0.49), teacher–student relationships (*M* = 3.22/3.31, *SD* = 0.75/0.72 in reading; *M* = 3.25/3.26, *SD* = 0.68/0.76 in mathematics; *M* = 3.21/3.12, *SD* = 0.67/0.85 in science) and academic achievements (*M* = 41.74/65.16, *SD* = 14.95/21.01 in reading; *M* = 42.94/52.86, *SD* = 15.17/18.70 in mathematics; *M* = 44.61/62.17, *SD* = 13.41/18.36 in science) at T1 and T2. The correlations results reflected that all variables at T1 had a positive association with corresponding variables at T2 (*r* = 0.22~0.58). In addition, SEC was positively related to teacher–student relationships and academic achievements across the three subjects at both T1 and T2 (*r* = 0.15~0.44). Teacher–student relationships and academic achievements were positively associated with each other at T1 and T2 in reading, mathematics and science (*r* = 0.17~0.26). Referring to the two indicators of peer relationships, SP had a much stronger correlation with SEC, teacher–student relationships and academic achievements across the three subjects at T1 and T2 than SI. 

In addition, the ANCOVA results (Table 1) showed the effects of students’ gender, age and their parents’ highest education on the corresponding differential score after controlling for the previous level of SEC, SP, SI, academic achievements and teacher–student relationships in reading, mathematics and science. The two-way and three-way interactions of students’ gender, age and their parents’ highest education had significant effects on the differential scores of SEC, SI, reading achievement, mathematics achievement and reading teacher–student relationship. Thus, students’ age, gender and their parents’ highest education were controlled in the following analysis.

### 3.3. Attrition Analysis

Approximately 30.19% of participants were missing in the second survey. The problem of longitudinal attrition was assessed using an independent *t* test. The results showed that there were no statistically significant differences between completers and noncompleters in gender [*t*_(3984)_ = 2.02, *p* > 0.05], age [*t*_(3012)_ = 0.05, *p* > 0.05], highest education of parents [*t*_(3727)_ = 0.08, *p* > 0.05], science achievement [*t*_(3940)_ = 0.10, *p* > 0.05] or teacher–student relationships [*t*_(3645)_ = 1.63, *p* > 0.05 in reading; *t*_(3738)_ = 1.19, *p* > 0.05 in mathematics; *t*_(3790)_ = 0.02, *p* > 0.05 in science]. However, the statistically significant differences found in reading achievement [*t*_(3928)_ = 8.32, *p* < 0.05, d = 0.10] and mathematics achievement [*t*_(3941)_ = 3.95, *p* < 0.05, d = 0.07] were relatively small. The missing data were processed with the full information maximum likelihood (FIML) method, which can produce unbiased and effective parameter estimates [56]. 

### 3.4. Measurement Invariance Test

The results of the measurement invariance tests are shown in Table 2. Absolute differences were calculated by subtracting the fit indices of the less strictly limited model from those of the more strictly limited model. The differences in the goodness-of-fit indices (CFI, TLI and RMSEA) between the configural and metric invariance model and between metric and scalar invariance model were smaller than 0.01. The results provided support for a strong invariance model for SEC and teacher–student relationships in reading, mathematics and science.

### 3.5. Analysis of the Cross-Lagged Paths

The three models demonstrated a good fit to the data (χ^2^_(284)_ = 694.21, *p* < 0.001, CFI = 0.983, TLI = 0.977, RMSEA = 0.019, SRMR = 0.019 in reading; χ^2^_(284)_ = 700.481, *p* < 0.001, CFI = 0.983, TLI = 0.978, RMSEA = 0.019, SRMR = 0.020 in mathematics; χ^2^_(284)_ = 684.03, *p* < 0.001, CFI = 0.984, TLI = 0.979, RMSEA = 0.019, SRMR = 0.019 in science). The prediction coefficients of the paths in the three models are shown in Figure 1, Figure 2 and Figure 3, and insignificant paths have not been drawn.

In the reading model (Figure 1), the variables at T1 were correlated, except for SI with SP and reading teacher–student relationship. At T2, the correlations between SI and SEC, reading teacher–student relationship and reading achievement were not significant (*p* > 0.05). The standardized autoregressive coefficients (see Figure 1 upper) showed that SEC (β = 0.30, *p* < 0.001), SP (β = 0.50, *p* < 0.001), SI (β = 0.34, *p* < 0.001), reading teacher–student relationship (β = 0.21, *p* < 0.001) and reading achievement (β = 0.41, *p* < 0.001) were stable between the two waves. The cross-lagged effects revealed reciprocal associations between reading achievement and SEC (β = 0.14/0.19, *p* < 0.001), reading teacher–student relationship (β = 0.12/0.07, *p* < 0.001) and SP (β = 0.08/0.10, *p* < 0.001); between SEC and reading teacher–student relationship (β = 0.09/0.11, *p* < 0.001); as well as between SP and SI (β = −0.10/−0.07, *p* < 0.001). In addition, the effects of prior SP on later SEC (β = 0.06, *p* < 0.01) and prior reading achievement on later SI (β = 0.06, *p* < 0.01) were significant.

In the mathematics model (Figure 2), the correlations between SI with SP, mathematics teacher–student relationship and mathematics achievement were not significant (*p* > 0.05) at T1. The correlations between SI and SEC, mathematics teacher–student relationship and mathematics achievement were not significant (*p* > 0.05) at T2. The standardized autoregressive coefficients (see Figure 1 middle) showed that SEC (β = 0.29, *p* < 0.001), SP (β = 0.50, *p* < 0.001), SI (β = 0.34, *p* < 0.001), mathematics teacher–student relationship (β = 0.13, *p* < 0.001) and mathematics achievement (β = 0.49, *p* < 0.001) were stable. The cross-lagged effects revealed reciprocal associations between mathematics achievement and SEC (β = 0.17/0.15, *p* < 0.001), mathematics teacher–student relationship (β = 0.16/0.05, *p* < 0.05) and SP (β = 0.06/0.11, *p* < 0.001); between SEC with mathematics teacher–student relationship (β = 0.08/0.10, *p* < 0.001); as well as between SP with SI (β = −0.09/−0.06, *p* < 0.001). In addition, the effects of prior SP on later SEC (β = 0.05, *p* < 0.01) and mathematics teacher–student relationship (β = 0.05, *p* < 0.01) and of prior SEC on later SI (β = 0.06, *p* < 0.01) were significant.

In the science model (Figure 3), the correlations between SI and SP, science teacher–student relationship and science achievement were not significant at T1 (*p* > 0.05). The correlations between SI and SEC, science teacher–student relationship and science achievement were not significant (*p* > 0.05). The standardized autoregressive coefficients (see Figure 1 lower) showed that SEC (β = 0.31, *p* < 0.001), SP (β = 0.50, *p* < 0.001), SI (β = 0.34, *p* < 0.001), science teacher–student relationship (β = 0.21, *p* < 0.001) and science achievement (β = 0.45, *p* < 0.001) were stable. Reciprocal associations between science achievement and SEC (β = 0.13/0.12, *p* < 0.001) and SP (β = 0.07/0.08, *p* < 0.001) and between SP and SI (β = −0.09/−0.06, *p* < 0.01) were revealed. In addition, the effects of prior science teacher–student relationship (β = 0.10, *p* < 0.001) and SP (β = 0.06, *p* < 0.001) on later SEC were significant. The effect of prior science achievement on later science teacher–student relationship was significant (β = 0.11, *p* < 0.001).

## 4. Discussion

This study empirically analyzed the reciprocal associations among students’ SEC, peer relationships and teacher–student relationships and academic achievements. SP and SI are the two indicators of peer relationships. The patterns of these reciprocal relationships in three subjects (reading, mathematics and science) were explored. In this study, the following major research findings were discovered. It revealed the more realistic relationships among these variables, which improved the ecological validity of the findings.

### 4.1. Findings and Discussions

First, this reciprocal association between SEC and academic achievements was consistent in the three subjects, supporting H1. Many SEC skills taught in school, such as self-control and cooperation [63], impact students’ academic experiences [64]. Furthermore, students’ academic achievements were predictive of their subsequent SEC. Usually, the feedback of academic achievements provides valuable insights for children to make adjustments [30]. It is beneficial for students to improve their SEC. The reciprocal association between SEC and academic achievements posits that students’ SEC can facilitate their learning and that their academic achievements can impact their SEC.

Second, SEC and teacher–student relationships in reading and mathematics had bidirectional relationships, partially supporting H2. This indicates that teacher–student relationships are necessary to promote the development of students’ SEC [12,65]. And SEC empowers children to proactively manage their emotions and behaviors, thus establishing close connections with their teachers [11,12,66]. In addition, SP positively predicted SEC in the three subjects, partially supporting H3. Students who are preferred by peers tend to develop stronger social competence [51], while SEC did not predict subsequent SP in the three subjects. This might be due to the stability of peer relationships, which make the SP not susceptible to impact [67]. Furthermore, the prediction of SEC on SI was only significant in the mathematics model, but the effect size was small and deserved further validation. The difference between the effects of SP and SI suggests that they are independent structures of peer relationships [35]. It also provides evidence that being genuinely well-liked by peers significantly influences the development of SEC, rather than being seen as popular but not necessarily well-liked. 

Third, there were reciprocal associations between teacher–student relationships and academic achievements in reading and mathematics, partially supporting H4. Close relationships with teachers provide students with emotional support and guidance, and equip them with the necessary knowledge and skills to excel academically [17,27]. High academic achievements also contribute to the establishment of positive teacher–student relationships. The reciprocal relationships between SP and academic achievements in the three subjects, partially supporting H5. This showed that positive peer relationships benefited students’ academic achievements [68] and receiving more peer preference led to more learning engagement and increased performance [69]. Moreover, only the prediction of reading achievement on SI was significant, and its small effect size makes further validation necessary. The difference between the effects of SP and SI implies that being genuinely well-liked by peers is positively correlated with academic achievements compared to the visibility among peers. 

Fourth, some differences were identified among the reading, mathematics and science models. Specifically, the reciprocal associations between teacher–student relationship with SEC and academic achievement were insignificant in science. This may reflect the weak role of the science teacher–student relationship. Insufficient science teachers in rural areas means science is often taught by reading or mathematics teachers [27]. In this case, the science teacher–student relationship comprises students’ relationships with the part-time science teachers. Additionally, the predictions were observed between SEC and later SI in mathematics, as well as between reading achievement and later SI. Moreover, SP positively predicted later mathematics teacher–student relationships, highlighting the positive effect of peer preference [21,22,70]. While these effects offer fresh insights into the associations between SI and SEC and reading achievement, as well as between SP and mathematics teacher–student relationship, the instability and small effect sizes necessitate further investigation.

### 4.2. Theoretical and Practical Implications

In general, this study provides a more comprehensive understanding of the directions and associations among SEC, SP, SI, teacher–student relationships and academic achievements in reading, mathematics and science. In addition, the study considered two indicators of peer relationships, namely SP and SI and found that being genuinely well-liked by peers had higher levels of correlation with the development of SEC and academic achievements. This lays the basis for considering different forms of peer relationships in the future [34,35]. Furthermore, the study examined the feedback effect of academic achievements on SEC and interpersonal relationships. The study enriches the literature on the feedback effect of academic achievements. Finally, by synthesizing the associations among these factors in reading, mathematics and science, the results and conclusions may be highly credible.

The practical implications of the results point toward strategies for enhancing students’ SEC, academic achievements and the establishment of positive peer and teacher–student relationships. First of all, teachers were the attachments of students in school, providing them with care and support. Fostering close teacher–student relationships can be instrumental in promoting students’ SEC and facilitating their academic success. In the process of building positive teacher–student relationships, teachers are required to manage existing and potential teacher–student conflicts properly. When students are experiencing academic difficulties, teachers should provide counsel and advice in time. This could help them enhance their ability to solve difficulties, thereby avoiding the generation of emotional problems or problematic behaviors. Additionally, teachers need to provide timely feedback about their students’ progress and mistakes. Frequent and close communications are beneficial to the establishment of positive teacher–student relationships. 

Therefore, implementing SEL programs can be highly effective in improving students’ SEC, such as emotional regulation, problem-solving abilities and interpersonal skills. These skills and abilities are essential for students’ development. Thus, introducing and designing targeted SEL programs have the potential to cultivate students’ positive interpersonal relationships within the school context, ultimately leading to their improved academic achievements. It is necessary for schools to measure students’ SEC regularly to constantly monitor their development and changes in various aspects, which are the basis for timely adjusting and updating SEL programs. 

Additionally, the strong feedback effect of academic achievements was revealed in the current study. Given this, the role of academic achievements should be fully utilized in the education process. This raises requirements for educators that they are supposed to provide appropriate feedback for students promptly. In this instance, students could be informed of their performance, and develop a correct understanding and cognition about themselves. For instance, they can better understand which strategies are helpful to them and which strategies need improvement. In this case, students monitor their learning process and regulate learning strategies, which improves their adaptability and coping abilities. This further promotes the development of SEC and the establishment of good interpersonal relationships. 

Finally, the study revealed the relatively weak role of the science teacher–student relationship. This may be attributed to the mixture of characteristics of the science teacher–student relationship, resulting from the shortage of qualified science teachers. This implies inspiration for emphasis on science education, such as adding science teacher qualification tests and increasing reserves of science teachers.

### 4.3. Limitations and Outlooks

Although this study provides valuable insights, it is important to acknowledge several limitations of this study when interpreting the findings. The first limitation is that the study did not control for variables such as intelligence and prior cognitive ability, which might be associated with students’ SEC, academic achievements and positive interpersonal relationships [71,72]. The omission of these variables may have an influence on the results and should be considered in future studies. 

Furthermore, parent–child relationships are reported to have a significant impact on the growth and development of primary school students [73]. However, the study only focused on students’ interpersonal relationships in school and did not take parent–child relationships into account. This is not conducive to a comprehensive understanding of the role of interpersonal relationships among primary school students. In the future study, considering the effect of parent–child relationships could provide a more comprehensive understanding of the associations among students’ SEC, academic achievements and interpersonal relationships. 

Lastly, the study was based on two-wave data, which prevented the possibility of conducting longitudinal mediation analysis. And it is not appropriate to compare the results in the current study with previous mediation studies on the associations among SEC, interpersonal relationships and academic achievements. The cross-lagged panel design employed in this study has been criticized for confounding variance at both the inter- and intra-individual levels [74]. With only two time points, it is difficult to disentangle these effects. Future research could address this limitation by including more measurement points. Then, the investigation of intermediary variables and the disentanglement of inter- and intra-individual effects could be obtained.

## 5. Conclusions

In conclusion, this study revealed complex predictive associations among primary school students’ SEC, SP, SI, teacher–student relationships and academic achievements in reading, mathematics and science. Specifically, there were reciprocal associations between SEC with academic achievements and teacher–student relationships, as well as between academic achievements with SP and teacher–student relationships. In addition, the predictive relationships had some differences among the three subjects of reading, mathematics and science. The study provides a new understanding of the reciprocal associations among SEC, SP, SI, teacher–student relationships and academic achievements. It offers valuable directions for the enhancement of SEC, interpersonal relationships and academic achievements. Furthermore, by considering the complex interplay across the three subjects (reading, mathematics and science), the study offers specific implications for future subject-specific education.

## Figures and Tables

**Figure 1 behavsci-13-00922-f001:**
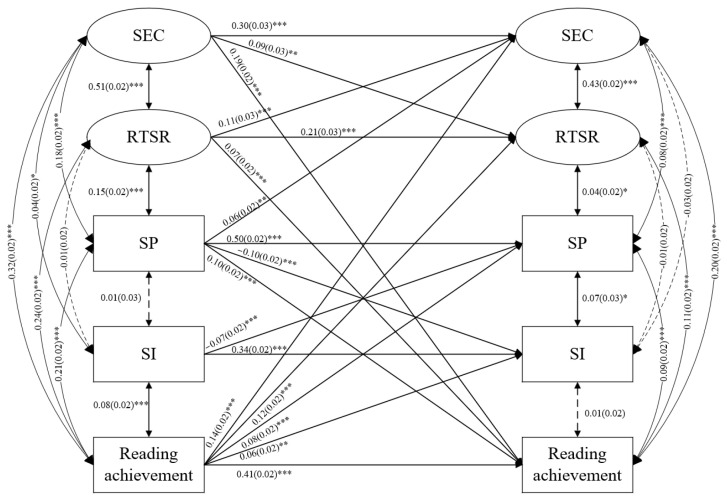
The reciprocal associations among SEC, SP, SI, teacher–student relationships and academic achievements in reading. * *p* < 0.05, ** *p* < 0.01, *** *p* < 0.001.

**Figure 2 behavsci-13-00922-f002:**
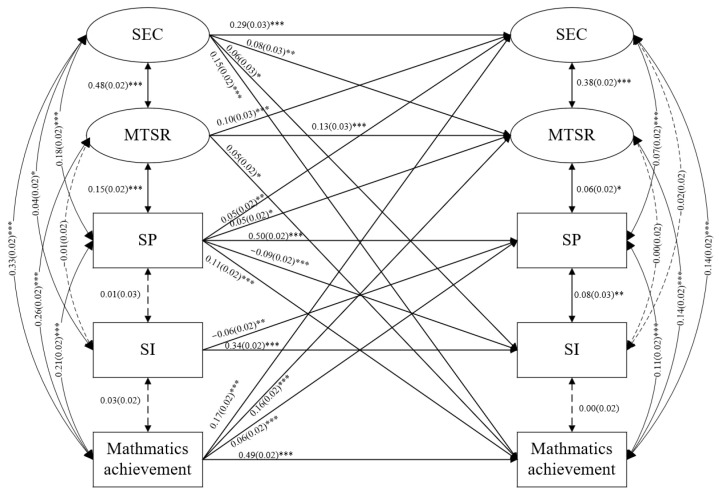
The reciprocal associations among SEC, SP, SI, teacher–student relationships and academic achievements in mathematics. * *p* < 0.05, ** *p* < 0.01, *** *p* < 0.001.

**Figure 3 behavsci-13-00922-f003:**
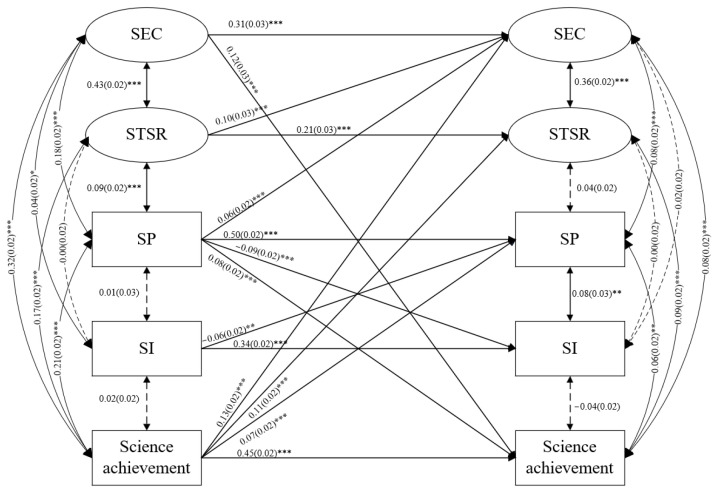
The reciprocal associations among SEC, SP, SI, teacher–student relationships and academic achievements in science. * *p* < 0.05, ** *p* < 0.01, *** *p* < 0.001.

**Table 1 behavsci-13-00922-t001:** ANCOVA results of gender, highest education of parents and age across SEC, SP, SI, academic achievements and teacher–student relationships in reading, mathematics and science.

Variable	DS of SEC	DS of SP	DS of SI	DS of Reading Achievement	DS of Mathematics Achievement	DS of Science Achievement	DS of RTSR	DS of MTSR	DS of STSR
Intercept	864.86 ***	0.02	0.02	238.29 ***	187.05 ***	228.55 ***	483.73 ***	372.24 ***	240.33 ***
Pretest	1187.68 ***	549.84 ***	908.19 ***	153.34 ***	276.84 ***	156.92 ***	1063.03 ***	778.85 ***	564.84 ***
Gender	1.04	4.05	0.23	0.17	0.67	6.81 **	1.77	0.09	0.34
Age	1.45	0.63	0.64	1.59	1.80	0.67	1.38	1.60	0.68
Education	1.29	1.40	3.38 **	1.25	0.74	0.99	2.22	023	0.69
Gender * Age	0.63	0.94	2.28 *	1.23	1.07	1.36	0.83	1.10	1.05
Gender * Education	0.38	1.42	1.54	3.29 *	2.46 *	1.70	3.26 *	1.71	2.23
Education * Age	2.00 **	1.05	1.66 *	0.79	1.03	1.15	1.78 *	1.40	1.03
Gender * Education * Age	0.85	0.53	2.03	1.81 *	0.99	1.57	1.26	0.56	1.16

* *p* < 0.05, ** *p* < 0.01, *** *p* < 0.001. DS means the differential score, obtained by subtracting the raw score of main variables at T2 from the corresponding raw score at T1. SEC is social–emotional competence. SP is social preference. SI is social impact. RTSR, MTSR and STSR are teacher–student relationships in reading, mathematics and science, respectively. Pretest refers to the SEC, SP, SI, academic achievements and teacher–student relationships in reading, mathematics and science at T1 according to the corresponding different outcome variables at T2. Education is the highest education of parents. The same as below.

**Table 2 behavsci-13-00922-t002:** The measurement invariance tests of SEC and teacher–student relationships in reading, mathematics and science.

Variable	Model	χ^2^	*df*	CFI	TLI	RMSEA	SRMR
SEC	1	623.04	96	0.958	0.942	0.041	0.028
	2	722.13	104	0.951	0.938	0.042	0.039
	3	852.04	112	0.941	0.931	0.045	0.041
	4	2864.28	118	0.813	0.791	0.079	0.140
RTSR	1	82.43	10	0.989	0.978	0.047	0.018
	2	140.45	14	0.981	0.972	0.053	0.051
	3	158.95	18	0.979	0.976	0.049	0.052
	4	633.11	23	0.907	0.919	0.090	0.112
MTSR	1	99.57	10	0.987	0.974	0.052	0.019
	2	137.89	14	0.982	0.974	0.052	0.041
	3	202.00	18	0.973	0.970	0.056	0.044
	4	609.72	23	0.915	0.926	0.088	0.107
STSR	1	92.07	10	0.990	0.980	0.050	0.015
	2	133.36	14	0.985	0.979	0.051	0.036
	3	178.33	18	0.980	0.978	0.053	0.044
	4	745.95	23	0.910	0.922	0.099	0.098

Models 1–4 are configural, metric, scalar and residual invariance models, respectively.

## Data Availability

The original data are available on reasonable request from the corresponding author.

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
