# Peer review of "Reciprocal Associations among Social–Emotional Competence, Interpersonal Relationships and Academic Achievements in Primary School"

_behavsci, 2023, doi:10.3390/bs13110922_

Round 1
Reviewer 1 Report
Comments and Suggestions for Authors
The study aimed to examine the reciprocal associations between social-emotional competence (SEC), interpersonal relationships (including teacher-student relationships and peer relationships), and academic achievement in reading, mathematics, and science of elementary school students.
The study has a large sample, and a longitudinal measurement with validated scales. As suggestions for improvement, it is recommended to reduce the introduction, half the text is enough, this is not a book chapter. Tables 1, 2, and 3 may be left over and presented in another format.The longitudinal analysis must be done with an ANCOVA, with the pretest as a covariate, and the categorical variable as the independent variable.
Thanks
Author Response
The study aimed to examine the reciprocal associations between social-emotional competence (SEC), interpersonal relationships (including teacher-student relationships and peer relationships), and academic achievement in reading, mathematics, and science of elementary school students.
The study has a large sample, and a longitudinal measurement with validated scales. As suggestions for improvement, it is recommended to reduce the introduction, half the text is enough, this is not a book chapter. Tables 1, 2, and 3 may be left over and presented in another format.
Response: Thank you very much for your suggestion. We have reduced the introduction section by deleting irrelevant content and redundant information. In addition, we have merged the Tables 1, 2, and 3 in the original manuscript into the Table 1 in the revised manuscript.
The longitudinal analysis must be done with an ANCOVA, with the pretest as a covariate, and the categorical variable as the independent variable.
Response: Thank you very much for your review of this manuscript and suggestion. We have added the content about analysis of covariance. The corresponding results and description are provided in the section “3.2. Descriptive statistics, correlations and ANCOVA” (line 258-269). Thank you again for your valuable suggestion.
Reviewer 2 Report
Comments and Suggestions for Authors
The study provides valuable insights into the reciprocal associations among social-emotional competence (SEC), interpersonal relationships (including teacher-student relationships and peer relationships), and academic achievements in primary school students. One of its strengths lies in the extensive sample size, with 3995 fourth-grade students participating in the initial survey, and 2789 of them tracked in a follow-up survey two years later. This large sample size enhances the study's robustness and generalizability of findings. Another notable strength is the utilization of structural equation modeling to investigate cross-lagged relationships among variables across time. This statistical approach allows for a comprehensive analysis of the complex interplay between SEC, interpersonal relationships, and academic achievements. The study's findings regarding the reciprocal associations between these factors offer valuable insights for educators and policymakers seeking to enhance students' overall development. Furthermore, the study's acknowledgment of variations in these reciprocal associations across different subjects, such as reading, mathematics, and science, adds depth to the research. It highlights the importance of considering subject-specific education strategies to cater to the unique dynamics within each academic domain. Overall, this study's comprehensive approach and insightful findings contribute significantly to the understanding of how SEC, interpersonal relationships, and academic achievements are interrelated in primary school students.
Author Response
The study provides valuable insights into the reciprocal associations among social-emotional competence (SEC), interpersonal relationships (including teacher-student relationships and peer relationships), and academic achievements in primary school students. One of its strengths lies in the extensive sample size, with 3995 fourth-grade students participating in the initial survey, and 2789 of them tracked in a follow-up survey two years later. This large sample size enhances the study's robustness and generalizability of findings. Another notable strength is the utilization of structural equation modeling to investigate cross-lagged relationships among variables across time. This statistical approach allows for a comprehensive analysis of the complex interplay between SEC, interpersonal relationships, and academic achievements. The study's findings regarding the reciprocal associations between these factors offer valuable insights for educators and policymakers seeking to enhance students' overall development. Furthermore, the study's acknowledgment of variations in these reciprocal associations across different subjects, such as reading, mathematics, and science, adds depth to the research. It highlights the importance of considering subject-specific education strategies to cater to the unique dynamics within each academic domain. Overall, this study's comprehensive approach and insightful findings contribute significantly to the understanding of how SEC, interpersonal relationships, and academic achievements are interrelated in primary school students.
Response: Thank you very much for your review of this manuscript and for your approval of this study. We will continue to pay attention to this research field and conduct subsequent research. And we are looking forward to further communication and discussion with you. Thank you very much.
Reviewer 3 Report
Comments and Suggestions for Authors
Several aspects of the SEC are considered within the concept of hot executive functions, which are important for learning processes. Therefore, I suggest including a brief description of this relationship. Additionally, it is a very thorough approach to the effect of these abilities in children learning process and interpersonal relationships.
Author Response
Several aspects of the SEC are considered within the concept of hot executive functions, which are important for learning processes. Therefore, I suggest including a brief description of this relationship. Additionally, it is a very thorough approach to the effect of these abilities in children learning process and interpersonal relationships.
Response: Thank you very much for your suggestion. Through literature review, we have learned that incorporating motivational and emotional components (such as rewards) during the execution of target tasks is called hot execution function. It is usually measured by experimental tasks. Moreover, preschool age is the important period for children to develop their hot executive function. Although academic emotions and motivation are highly correlated with important variables such as social emotional competence (SEC), which was one of the main studying variables in this study. SEC is not hot executive functions. But we can clarify their differences and connections in our subsequent research. Your suggestion provides us with a certain direction for future research. Thank you again for your review and suggestion.
References:
Castillo, A., & Lopez, L. D. (2022). Studying hot executive function in infancy: Insights from research on emotional development. Infant Behavior and Development, 69, 101773. https://doi.org/10.1016/j.infbeh.2022.101773
Enke, S., Gunzenhauser, C., Johann, V. E., Karbach, J., & Saalbach, H. (2022). “Hot” executive functions are comparable across monolingual and bilingual elementary school children: Results from a study with the Iowa Gambling Task. Frontiers in Psychology, 13, 988609. https://doi.org/10.3389/fpsyg.2022.988609
Reviewer 4 Report
Comments and Suggestions for Authors
Thank you for the opportunity to review this manuscript “Reciprocal associations among social-emotional competence, interpersonal relationships and academic achievements in primary school.” The manuscript examined the associations between social-emotional competence, interpersonal relationships, and academic achievements based on surveys conducted at two different times. Below are my suggestions for revisions before the manuscript can be published.
1. The introduction overall is well-written. The authors provided the rationale of why the topic should be further explored. However, some points require further clarification.
The sentence “Students’ success in school is not only influenced by cognitive abilities and opportunities” What do opportunities mean? There is no context to describe opportunities. Can the authors clarify what the opportunities mean here?
While reading the initial introduction, I noticed that the authors abruptly introduced the ecological theory without any prior mention or context. Not just the ecological theory, the authors also mentioned other theories, such as social cognitive theory and attachment theory. I would suggest removing those theories in this entire manuscript for several reason.
· Based on my reading, I do not think the authors “apply” a theory as a framework to examine the hypotheses.
· The theories mentioned can be confusing for readers, especially if these theories aren't the primary focus of the manuscript. Are these theories the primary focus? If not, I wonder why the authors chose to introduce them. If they are the primary focus, the authors should utilize these theories as frameworks to restructure and refine the manuscript. However, I think the authors did not intend to use the theories as a framework. For instance, the social cognitive theory and attachment theory are only briefly mentioned. Moreover, based on the data analysis methods described, it doesn't appear that the authors have consistently applied these theories as guiding frameworks.
· It seems the authors mentioned theories to justify the associations among social-emotional competence, interpersonal relationships and academic achievements. While their intention seems to be to use these theories as a bridge to justify the associations, I believe a strong rationale for the associations can be provided even without these theories. The extensive previous literature already serves as sufficient justification for these associations, which makes the additional theoretical information redundant.
The authors should offer a more detailed explanation of social preference (SP) and social impact (SI) as they appear important variables to the manuscript, especially given that SP is addressed within the hypotheses. Currently, the manuscript only briefly introduced the concepts of SP and SI. Once again, I suggest a reconstruction of the introduction. Redundant details, such as theories, could be removed, while emphasizing and elaborating on crucial elements such as SP and SI.
“Cillessen and Rose advocated to think different forms of peer relationships.” What does it mean? The description is not clear.
The authors introduced SI by saying “SI reflects the degree of being neglected or controversial among peers.” Is it correct introduction? Is it SI all about? Out of curiosity, I took the initiative to research SI myself, and my findings suggest a different understanding of the concept. The authors should have an exclusive section to introduce SP and SI. The information on SP and SI in the manuscript is too brief and might lead to misunderstandings.
“Due to the little research exploring the effect of SI, the association between SI
and SEC would be explored in the current study.” But there is no hypothesis for
the association between SI and SEC. Is it a typo? I only see a hypothesis for SP.
This is even more confusing “The relationships between SI and academic
achievements in reading, mathematics and science were investigated and not be
hypothesized in this study due to the little research exploring the effect of
SI.” Why did the authors decide not to explore the effect of SI “due to the little
research?” Previously, the authors just mentioned “due to the little research
exploring the effect of SI, the association between SI and SEC would be
explored in the current study” This is very confusing.
2. For the methods section, please review the following suggestions:
Please add participant recruitment. Did the authors collaborate with schools to recruit participants? Or how?
In the manuscript, the authors only specify the timing for the academic achievement tests and the questionnaires. I'm curious about the timing for other measures, such as those assessing social-emotional competence, teacher-student relationships, and peer relationships.
Regarding the measures, it appears that all of them were administered in their Chinese versions. Is the provided reliability information for Chinese versions? Were these Chinese versions pre-existing, or were they translated by the authors for the purpose of this study? Additionally, is there any information available on the validity of these Chinese versions?
3. For the discussion and implications sections, I recommend that the authors first restructure the introduction and then ensure that the discussion directly addresses the points raised in the introduction. Furthermore, a more in-depth exploration of SP and SI would greatly benefit readers
The authors should adopt a more holistic approach when presenting practical implications. As an educator, I found the current practical implications confusing. Given the interconnectedness of social-emotional competence, interpersonal relationships, and academic achievements, it would be beneficial if the authors could frame the implications more broadly. They should also offer straightforward suggestions for educators. By doing so, educators would use strategies, as recommended by the authors, to enhance students' social-emotional competence, interpersonal relationships, and academic performances.
Author Response
Thank you for the opportunity to review this manuscript “Reciprocal associations among social-emotional competence, interpersonal relationships and academic achievements in primary school.” The manuscript examined the associations between social-emotional competence, interpersonal relationships, and academic achievements based on surveys conducted at two different times. Below are my suggestions for revisions before the manuscript can be published.
- The introduction overall is well-written. The authors provided the rationale of why the topic should be further explored. However, some points require further clarification.
The sentence “Students’ success in school is not only influenced by cognitive abilities and opportunities” What do opportunities mean? There is no context to describe opportunities. Can the authors clarify what the opportunities mean here?
Response: Thank you very much for your question and suggestion. We did not mention clearly what opportunities are in the original manuscript, which might confuse the readers. The opportunities refer to students’ opportunities to learn (OTL), including available resources, such as curriculum documents and qualified teachers. To avoid confusion, we have deleted the content about opportunities.
While reading the initial introduction, I noticed that the authors abruptly introduced the ecological theory without any prior mention or context. Not just the ecological theory, the authors also mentioned other theories, such as social cognitive theory and attachment theory. I would suggest removing those theories in this entire manuscript for several reason.
Based on my reading, I do not think the authors “apply” a theory as a framework to examine the hypotheses.
The theories mentioned can be confusing for readers, especially if these theories aren't the primary focus of the manuscript. Are these theories the primary focus? If not, I wonder why the authors chose to introduce them. If they are the primary focus, the authors should utilize these theories as frameworks to restructure and refine the manuscript. However, I think the authors did not intend to use the theories as a framework. For instance, the social cognitive theory and attachment theory are only briefly mentioned. Moreover, based on the data analysis methods described, it doesn't appear that the authors have consistently applied these theories as guiding frameworks.
It seems the authors mentioned theories to justify the associations among social-emotional competence, interpersonal relationships and academic achievements. While their intention seems to be to use these theories as a bridge to justify the associations, I believe a strong rationale for the associations can be provided even without these theories. The extensive previous literature already serves as sufficient justification for these associations, which makes the additional theoretical information redundant.
Response: Thank you very much for your question and suggestion. Based on your suggestions displayed above, we have removed the corresponding description and explanation about theories in the original manuscript, such as ecological system theory, social cognitive theory and attachment theory. The revised manuscript is more concise and clearer.
The authors should offer a more detailed explanation of social preference (SP) and social impact (SI) as they appear important variables to the manuscript, especially given that SP is addressed within the hypotheses. Currently, the manuscript only briefly introduced the concepts of SP and SI. Once again, I suggest a reconstruction of the introduction. Redundant details, such as theories, could be removed, while emphasizing and elaborating on crucial elements such as SP and SI.
The authors introduced SI by saying “SI reflects the degree of being neglected or controversial among peers.” Is it correct introduction? Is it SI all about? Out of curiosity, I took the initiative to research SI myself, and my findings suggest a different understanding of the concept. The authors should have an exclusive section to introduce SP and SI. The information on SP and SI in the manuscript is too brief and might lead to misunderstandings.
Response: Thank you very much for your suggestion. We have removed minor content and redundant information related to this study. And we clarified the concepts and meanings of social preferences and social influence in the section “1.2. Social-emotional competence and interpersonal relationships” (line 88-97).
“Cillessen and Rose advocated to think different forms of peer relationships.” What does it mean? The description is not clear.
Response: Thank you so much for your question and suggestion. Although social preference (SP) and social impact (SI) might have the same external manifestations, namely peer popularity, but their internal characteristics are different. The former represents that children are genuinely well-liked by their peers, while the latter reflects that children are seen as popular but are not necessarily well-liked. Thus, researchers believe that different forms of peer relationships should be considered. We clarified the description about the sentence in the section “1.2. Social-emotional competence and interpersonal relationships” (line 89-92).
“Due to the little research exploring the effect of SI, the association between SI and SEC would be explored in the current study.” But there is no hypothesis for the association between SI and SEC. Is it a typo? I only see a hypothesis for SP.
This is even more confusing “The relationships between SI and academic achievements in reading, mathematics and science were investigated and not be hypothesized in this study due to the little research exploring the effect of SI.” Why did the authors decide not to explore the effect of SI “due to the little research?” Previously, the authors just mentioned “due to the little research exploring the effect of SI, the association between SI and SEC would be explored in the current study” This is very confusing.
Response: Thank you very much for your question. We have revised the content and references in the introduction section by removing unrelated content and adding description about social impact (SI). The research hypotheses we proposed in the manuscript were based on previous research results. Based on previous findings on the relationship between SI and aggressive behaviors as well as the relationship between SI and academic achievement, we may consider SI to be related to low SEC and academic achievements. Therefore, we have adjusted the hypotheses in the introduction section (line 109-113, line 135-142).
- For the methods section, please review the following suggestions:
Please add participant recruitment. Did the authors collaborate with schools to recruit participants? Or how?
Response: Thank you for your suggestion. We added the recruitment information in section “2.1. Participants and procedure” (line 147-151).
In the manuscript, the authors only specify the timing for the academic achievement tests and the questionnaires. I'm curious about the timing for other measures, such as those assessing social-emotional competence, teacher-student relationships, and peer relationships.
Response: Thank you very much for your question and suggestion. The questionnaires included measures of social-emotional competence, teacher-student relationships, and peer relationships. These were the individual information of students. In addition, the measurement items of the highest level of education of parents were also included in the questionnaires, which represent students' family background information. As the background questionnaires are not achievement tests, there is no specific completion time specified. Therefore, we made revisions in the revised manuscript to make the components of the questionnaire clear (line 156-159).
Regarding the measures, it appears that all of them were administered in their Chinese versions. Is the provided reliability information for Chinese versions? Were these Chinese versions pre-existing, or were they translated by the authors for the purpose of this study? Additionally, is there any information available on the validity of these Chinese versions?
Response: Thank you very much for your question and suggestion. All of measures were administered in Chinese. The measure of social emotional competence, namely the Chinese version of the Delaware Social and Emotional Competency Scale (DSECS-SCV) is the Chinese version. Teacher-student relationship scale was translated from PISA. The wording was modified to make these items suitable for fourth-grade and sixth-grade students in China. In the “2.2. Measures” section, we added the information about the validity of the DSECS-SCV in its original published article and the construct validity of the teacher-student relationship scale in our study (line 170-186).
- For the discussion and implications sections, I recommend that the authors first restructure the introduction and then ensure that the discussion directly addresses the points raised in the introduction. Furthermore, a more in-depth exploration of SP and SI would greatly benefit readers。
Response: Thank you very much for your suggestion. We have made modifications to the introduction section by removing theoretical content and redundant information related to researches. We also have added the meaning of social preference and social impact, as well as descriptions of their relevant researches. In addition, we have reorganized the discussion section based on the logic of the introduction and the proposed research hypotheses.
The authors should adopt a more holistic approach when presenting practical implications. As an educator, I found the current practical implications confusing. Given the interconnectedness of social-emotional competence, interpersonal relationships, and academic achievements, it would be beneficial if the authors could frame the implications more broadly. They should also offer straightforward suggestions for educators. By doing so, educators would use strategies, as recommended by the authors, to enhance students' social-emotional competence, interpersonal relationships, and academic performances.
Response: Thank you very much for your suggestion. In the section of implications, we have compressed the content of theoretical implications and added the content of practical implications (line 411-457).
Round 2
Reviewer 1 Report
Comments and Suggestions for Authors
The study aimed to examine the reciprocal associations between social-emotional competence (SEC), interpersonal relationships (including teacher-student relationships and peer relationships), and academic achievement in reading, mathematics, and science of elementary school students.
The study is well planned, the bibliographic review is adequate, and so is the methodology, with a very large sample.
My only doubt is in the Attrition analysis, the t test, I do not consider it optimal. An ancova of the differential score of the posttest, minus the pretest, is more appropriate, with the preters as a covarial.
Table 1, I don't see much sense, maybe a summary is better
Table 2 is not understood, I refer to my way of understanding the ancova with differential scores.
Author Response
Response: Thank you very much for your suggestion. We removed Table 1 and reported the main results in the “3.2. Descriptive statistics, correlations and ANCOVA” section.
Moreover, we corrected the content about the ANCOVA by changing the outcome variable from the raw scores of the main variables at T2 to the differential scores of the main variables between T1 and T2.
However, referring to the attrition analysis, we did not have access to the differential scores of the missing participating students, prevents us from conducting ANCOVA of the differential scores to test attrition bias. Through literature review, we found that t-test was one of the most common methods for detecting attrition bias in the characteristics of the sample (Miller & Wright, 1995; Nicholson et al., 2017). It aims to compare those participants who responded to all waves of the study with those who dropped out after only one wave. And the way that comparing means of important variables from the first wave between the missing and retained participants was used in many studies, to determine if the differences are statistically significantly different (e.g., Bono et al., 2007; Lieneman et al., 2020; Rübsamen et al., 2017; Sengupta & Gupta, 2012; Steinhausen et al., 2020). Thus, we choose conduct t-tests to test the attrition bias in our study.
Thank you again for your suggestion.
References:
Bono, C., Ried, L. D., Kimberlin, C., & Vogel, B. (2007). Missing data on the center for epidemiologic studies depression scale: a comparison of 4 imputation techniques. Research in Social and Administrative Pharmacy, 3(1), 1-27. https://doi.org/10.1016/j.sapharm.2006.04.001
Lieneman, C. C., Girard, E. I., Quetsch, L. B., & McNeil, C. B. (2020). Emotion regulation and attrition in parent–child interaction therapy. Journal of child and family studies, 29, 978-996. https://doi.org/10.1007/s10826-019-01674-4
Miller, R. B., & Wright, D. W. (1995). Detecting and correcting attrition bias in longitudinal family research. Journal of Marriage and the Family, 921-929. https://doi.org/10.2307/353412
Nicholson, J. S., Deboeck, P. R., & Howard, W. (2015). Attrition in developmental psychology: a review of modern missing data reporting and practices. International Journal of Behavioral Development, 41(1), 143–153. https://doi.org/10.1177/0165025415618275
Rübsamen, N., Akmatov, M. K., Castell, S., Karch, A., & Mikolajczyk, R. T. (2017). Factors associated with attrition in a longitudinal online study: results from the HaBIDS panel. BMC medical research methodology, 17(1), 1-11. https://doi.org/10.1186/s12874-017-0408-3
Sengupta, S., & Gupta, A. (2012). Exploring the dimensions of attrition in Indian BPOs. The International Journal of Human Resource Management, 23(6), 1259-1288. https://doi.org/10.1080/09585192.2011.561211
Steinhausen, H. C., Spitz, A., Aebi, M., Metzke, C. W., & Walitza, S. (2020). Selective attrition does not affect cross-sectional estimates of associations with emotional and behavioral problems in a longitudinal study with onset in adolescence. Psychiatry research, 284, 112685. https://doi.org/10.1016/j.psychres.2019.112685